# Alterations in Brain Cannabinoid Receptor Levels Are Associated with HIV-Associated Neurocognitive Disorders in the ART Era: Implications for Therapeutic Strategies Targeting the Endocannabinoid System

**DOI:** 10.3390/v13091742

**Published:** 2021-08-31

**Authors:** Mary K. Swinton, Erin E. Sundermann, Lauren Pedersen, Jacques D. Nguyen, David J. Grelotti, Michael A. Taffe, Jennifer E. Iudicello, Jerel Adam Fields

**Affiliations:** Department of Psychiatry, University of California, San Diego, CA 92093, USA; mary.swinton@western.edu (M.K.S.); esundermann@health.ucsd.edu (E.E.S.); lauren.pedersen@louisville.edu (L.P.); j5nguyen@health.ucsd.edu (J.D.N.); dgrelotti@health.ucsd.edu (D.J.G.); mtaffe@health.ucsd.edu (M.A.T.); jiudicello@health.ucsd.edu (J.E.I.)

**Keywords:** cannabinoid receptor, inflammation, astrocytes, immunohistochemistry

## Abstract

HIV-associated neurocognitive disorders (HAND) persist despite the advent of antiretroviral therapy (ART), suggesting underlying systemic and central nervous system (CNS) inflammatory mechanisms. The endogenous cannabinoid receptors 1 and 2 (CB_1_ and CB_2_) modulate inflammatory gene expression and play an important role in maintaining neuronal homeostasis. Cannabis use is disproportionately high among people with HIV (PWH) and may provide a neuroprotective effect for those on ART due to its anti-inflammatory properties. However, expression profiles of CB_1_ and CB_2_ in the brains of PWH on ART with HAND have not been reported. In this study, biochemical and immunohistochemical analyses were performed to determine CB_1_ and CB_2_ expression in the brain specimens of HAND donors. Immunoblot revealed that CB_1_ and CB_2_ were differentially expressed in the frontal cortices of HAND brains compared to neurocognitively unimpaired (NUI) brains of PWH. CB_1_ expression levels negatively correlated with memory and information processing speed. CB_1_ was primarily localized to neuronal soma in HAND brains versus a more punctate distribution of neuronal processes in NUI brains. CB_1_ expression was increased in cells with glial morphology and showed increased colocalization with an astroglial marker. These results suggest that targeting the endocannabinoid system may be a potential therapeutic strategy for HAND.

## 1. Introduction

Human immunodeficiency virus (HIV) in the antiretroviral therapy (ART) era has transformed from a terminal illness to a chronic disease with a life expectancy approaching that of seronegative patients [1]. However, even despite effective virologic suppression with ART, HIV-associated neurocognitive disorders (HAND) persist, affecting up to 50% of people with HIV (PWH) [2]. HIV is known to seed the brain within days of infection, and while ART has proven effective at suppressing viral loads and reducing progression to acquired immunodeficiency syndrome (AIDS), it does not eradicate central nervous system (CNS) viral reservoirs [3,4]. Persistent low-level HIV replication, chronic inflammation, ART neurotoxicity, and aging comorbidities are thought to contribute to the neuropathogenesis of HAND [5,6,7,8]. Important avenues for future investigation will involve optimizing HIV therapy within the CNS.

The mechanisms driving HAND are likely multifactorial, but common prospective etiologies include disruptions in neuroinflammatory signaling and mitochondrial function [9], both pathways that are modulated by the endocannabinoid system (ECS) [10,11,12]. Indeed, impaired mitochondrial fission and fusion, dysregulated autophagy, premature apoptosis, and altered calcium homeostasis have all been implicated in the pathogenesis of HAND [9]. Postmortem brain studies of HAND decedents reveal persistent astrogliosis, microgliosis, inflammatory cytokines expression, and altered mitochondrial architecture [13,14,15]. Importantly, recent evidence has shown that the endocannabinoid system regulates neuronal and glial function in rodent brains [16]. Animal studies implicate the endocannabinoid system in brain functions that are commonly altered in patients with HAND, including learning and memory, executive function, and reinforcement behavior [16].

The ECS represents a promising therapeutic target for increasing resilience to HAND. Rates of cannabis (i.e., marijuana) use are disproportionately high in this population, with some studies showing as high as 77% of HIV-infected adults reporting marijuana use at least once in their lifetime compared to 44.5% in uninfected adults [17,18,19]. However, in the case of Shiau et al. [18], as with many surveys of a large cohort, the data may be subject to selection bias that skews the results. Nevertheless, many patients report that cannabis offsets symptoms of HIV and ART side effects, including neuropathic pain, nausea, myopathy, lipodystrophy, and mood problems [20,21]. Cannabis may have promising utility in treating various neurodegenerative disorders with underlying inflammatory processes, and the expression of cannabinoid receptors is associated with Alzheimer’s disease (AD) [22], multiple sclerosis (MS) [23], Huntington’s disease (HD) [24], and Down’s syndrome [25]. Cannabinoid receptors (CB) CB_1_ and CB_2_ are G protein coupled receptors [26]. The expression of CB_1_ and CB_2_ has been reported in HIV encephalitis [27], but the receptor expression in patients with a psychiatric diagnosis of HAND on ART remains unknown. A greater understanding of how cannabinoid receptor expression is associated with HAND is critical in evaluating the therapeutic potential of cannabinoid drugs, as well as their potential side effects. 

In this study, CB_1_ and CB_2_ receptor expression in postmortem brain specimens were investigated from a well-characterized cohort of HAND decedents on ART. CB_1_ and CB_2_ levels in the frontal cortex were assessed by immunoblot, and then CB_1_ cellular localization was investigated using immunohistochemical techniques. Lastly, CB_1_ and CB_2_ levels were correlated with clinical covariates. 

## 2. Materials and Methods

### 2.1. Study Population

Brain specimens from a total of 24 HIV+ donors were acquired from the National NeuroAIDS Tissue Consortium (NNTC) (Institutional Review Board [IRB] #080323) (Table 1). All studies were conducted in accordance with the code of ethics of the National Institutes of Health and the University of California, San Diego. Neuromedical and neuropsychological examinations were performed on each case within a median of 12 months prior to death. Exclusion criteria for subjects included a history of CNS opportunistic infections or diagnoses unrelated to HIV infection that might impact CNS functioning, such as neurologic, psychiatric, or metabolic disorders. The most common pathologies described were systemic cytomegalovirus (CMV), Kaposi sarcoma (HHV-8), and hepatic disease. A diagnosis of HAND was made according to a standardized evaluation, as described below [28].

### 2.2. Neuromedical and Neuropsychological Evaluation

All participants underwent a comprehensive neuromedical assessment that included a detailed medical history and structured set of examinations for detecting lifetime and current diagnoses [28,29]. For their baseline assessment, all subjects had venipuncture, cerebrospinal fluid (CSF), and urine samples collected. Clinical data plasma viral load (VL), postmortem interval, CD4 count, and neuropsychological measures were obtained for the HIV+ donor cohorts.

Neuropsychological evaluation for HAND diagnosis was performed across seven neurocognitive domains, including executive function, motor skill, processing speed, episodic memory, attention/working memory, language, and visual perception, as described by [28]. Raw test scores were transformed into normally-distributed T-scores adjusted for demographic variables, including age, education, gender, and race based on normative samples of HIV-participants and were then averaged across all tests to obtain a global cognitive T-score and within domains to obtain cognitive domain-specific T-scores [30]. Functional impairments in everyday life were assessed using the Lawton and Brody activities of daily living questionnaire [31] and patient’s assessment of own functional inventory (PAOFI) [32]. HAND classifications, i.e., asymptomatic neurocognitive impairment (ANI), mild neurocognitive disorder (MND), and HIV-associated dementia (HAD), were assigned based on participant responses to the everyday functioning questionnaires and performances on the neuropsychological test battery, according to the established criteria [2].

### 2.3. ImmunoBlot

Tissues from the white matter (WM) and gray matter (GM) of the frontal cortices dissected at autopsy from HAND and neurocognitively unimpaired (NUI) brains were homogenized and fractionated, as described by [33], using a buffer that promotes separation of membrane and cytosolic fractions (1.0 mmol/L HEPES (Thermo Fisher Scientific, Waltham, MA, USA, Gibco, cat. no. 15630-080), 5.0 mmol/L benzamidine, 2.0 mmol/L 2-mercaptoethanol (Gibco, cat. no. 21985), 3.0 mmol/L EDTA (MilliporeSigma, St. Louis MI, USA, Omni pur, cat. no. 4005), 0.5 mmol/L magnesium sulfate, 0.05% sodium azide; final pH 8.8). Human brain tissue samples (0.1 g) were homogenized in 0.7 mL of fractionation buffer (1.0 mM HEPES, 5.0 mM Benzaidine, 2.0 mM b-mercaptoethanol, 3.0 mM EDTA, 0.5 mM magnesium sulfate, 0.05% sodium azide, pH 8.8) containing phosphatase (MilliporeSigma cat# 524624) and protease inhibitor (MilliporeSigma cat# 539131) cocktails. The tissue homogenate was centrifuged at 5000× *g* for 5 min at room temperature, and the resulting supernatant was collected, placed in appropriate ultracentrifuge tubes, and centrifuged at 436,000× *g* for 1 h at 44 °C in a TL-100 rotor (Beckman Coulter, Brea, CA, USA). The supernatant was retained and represented the cytosolic fraction, and the pellets were resuspended in 0.2 mL of buffer and re-homogenized to obtain the membrane fraction.

After determination of the protein content of all samples by bicinchoninic acid assay (Thermo Fisher Scientific, cat. no. 23225), membrane fractions were resolved by SDS-PAGE and transferred onto PVDF membranes using the iBlot transfer system (Thermo Fisher Scientific: Invitrogen, cat. no. IB24001) and NuPage transfer buffer (ThermoFisher Scientific, cat. no NP0006). The membranes were incubated for 1 h in 5% bovine serum albumin blocking solution and phosphate-buffered saline-tween 20 (PBST). Membranes were then incubated overnight at 4 °C with primary antibodies against CB_1_ (Abcam, Cambridge, UK, cat. no. ab23703) and CB_2_ (Abcam, cat. No. ab3561). Importantly, both of these antibodies were validated by the manufacturer for specificity in CB_1_ and CB_2_ knockout mice. Following visualization, blots were stripped and probed with a mouse monoclonal antibody against β-actin (ACTB) (MilliporeSigma, cat. no. A5441) diluted 1:2000 in blocking buffer as a loading control. All blots were washed in PBST and then incubated with species-specific IgG conjugated to HRP (American Qualex, cat. no. A102P5) diluted 1:5000 in PBST and visualized with SuperSignal West Femto Maximum Sensitivity Substrate (Thermo Fisher Scientific, cat. no. 34096). Images were obtained, and semi-quantitative analysis was performed with the VersaDoc gel imaging system and Quantity One software (Bio-Rad version 4.3.0).

### 2.4. Correlational Analyses between CB_1_ and CB_2_ Protein Levels and Cognitive T-Scores, Viral Load, and CD4+ Cell Counts

We examined the relationship between global and domain-specific cognitive T-scores and clinical covariates, including plasma viral load and CD4+ cell counts and CB_1_ and CB_2_ protein expression separately using Pearson’s R correlations.

### 2.5. Immunohistochemistry and Double Immunofluorescence

Free-floating 40 μm thick vibratome sections of human brains were washed with PBS three times, pre-treated for 20 min in 3% H_2_O_2_, and blocked with 2.5% horse serum (Vector Laboratories, Burlingame, CA, USA, cat. no. S-2012) for 1 h at room temperature. Sections were incubated at 4 °C overnight with the primary antibody, CB_1_ (Abcam, cat. no. ab23703) diluted in PBS. Sections were then incubated in secondary antibody, Immpress HRP Anti-rabbit IgG (Vector Laboratories, cat. no. MP-7401) for 30 min, followed by NovaRED peroxidase (HRP) substrate made with NovaRED Peroxidase (HRP) Substrate Kit as per manufacturer’s instructions (Vector Laboratories, cat. no. SK-4800). Control experiments consisted of incubation with secondary antibody only. Tissues were mounted on Superfrost plus slides (Thermo Fisher Scientific, cat. no. 12-550-15) and coverslipped with cytoseal (Thermo Fisher Scientific, Richard Allen Scientific, cat. no. 8310-16). Immunostained sections were imaged with a digital Olympus microscope. For each case (*n* = 4 NUI and *n* = 4 HAND) a total of 3 sections (10 images per section at 200× magnification) were analyzed in order to quantify the average number of immunolabelled cells per field of view (400 μm^2^). CB_1_+ cells of glial morphology and pyramidal neuronal bodies were counted manually. The specimens were blind coded and then broken after quantification background levels were obtained in tissue sections immunostained in the absence of primary antibody. Unfortunately, due to technical difficulties with eliminating background signal, tissue sections were not amenable to immunohistochemistry with the CB_2_ antibody.

Double immunolabeling studies were performed to determine the percent colocalization of CB_1_ receptors with GFAP+ (astroglia) and MAP2+ (neurons) signal in frontal cortices, as described [13,34]. For this purpose, vibratome sections of human brains were immunostained with antibodies against CB_1_ with GFAP (Sigma–Aldrich, cat. no. G3893) for astrocytes and MAP2 (Santa Cruz Biotechnologies, cat# sc-32791) for neurons. Sections were then reacted with fluorescent secondary anti-bodies, goat anti-mouse IgG 488 (Invitrogen, cat. no. A11011) and goat anti-rabbit IgG 568 (Invitrogen, cat. no. A11036). Sections were mounted on Superfrost Plus slides and coverslipped with vectashield (Vector, cat. no. 1000). Sections were imaged with a Zeiss 63× (N.A. 1.4) objective on an Axiovert 35 microscope (Zeiss, San Diego, CA, USA) with an attached MRC1024 laser scanning confocal microscope system (Bio-Rad, Hercules, CA, USA). For each case (*n* = 4 NUI and *n* = 4 HAND) a total of 3 sections (10 images per section at 200× magnification) were analyzed in order to quantify the average number of immunolabelled cells per field of view (100 μm^2^). The percent colocalization was quantified using Image J and the SQUAASH method [35]. An examiner blinded to sample identification analyzed all immunostaining.

## 3. Results

### 3.1. CB_1_ and CB_2_ Expression Are Increased in HAND Brains on ART

To determine the expression levels of CB_1_ and CB_2_ in brains of HIV+ donors, we analyzed the frontal lobe lysates generated from WM and GM from HAND cases as well as NUI cases (Table 1). Brain lysate membrane fractions were analyzed for CB_1_, CB_2_, and ACTB levels by immunoblot. In brain lysates from WM from HAND cases, CB_1_ protein band intensity increased in MND and HAND compared to NUI and ANI (Figure 1A). The intensity of the band corresponding to CB_2_ was similar in all groups (Figure 1A). Densitometry analyses of bands for CB_1_ showed that protein levels were significantly increased ~2-fold in MND and HAD when compared to NUI and ANI (Figure 1B). Densitometry analysis of the band corresponding to CB_2_ revealed no significant difference between groups (Figure 1C). In brain lysates from GM from HAND cases, CB_1_ protein band intensity increased in ANI, MND, and HAND compared to NUI (Figure 1D). The intensity of the band corresponding to CB_2_ was less intense in ANI, MND, and HAND compared to NUI (Figure 1A). Densitometry analyses of bands for CB_1_ showed that protein levels significantly increased ~1.7-, 1.9-, and 2-fold in ANI, MND, and HAD, respectively, when compared to NUI (Figure 1E). Densitometry analysis of the band corresponding to CB_2_ revealed a significant reduction (~40%, 60%, and ~50%, respectively) in ANI, MND, and HAD, respectively, when compared to NUI between groups (Figure 1F). These results suggest that CB_1_ and CB_2_ expression levels are differentially altered GM in the brains of decedents that were diagnosed with HAND.

### 3.2. Elevated CB_1_ Expression in Brains of PWH Is Associated with Poorer Cognitive Function

As an exporatory analysis, and as data were available, CB_1_ and CB_2_ expression levels in GM and WM were correlated with global and cognitive domain-specific T-scores. CB_2_ expression levels in GM or WM did not relate to any cognitive outcome. Conversely, higher CB_1_ expression levels in GM significantly related to poorer memory T-scores (R = −0.45, *p* = 0.04; Figure 2A) and higher CB_1_ expression levels in WM significantly related to a poorer speed of information processing (R = −0.49, *p* = 0.03; Figure 2B).

### 3.3. Reduced CB_2_ Expression Is Associated with Increased Age

Age is an important factor in the development of neurocognitive dysfunction. As shown in Table 2, the NUI group had an average age approximately 13 years below the HAND groups. However, neurocognitive T-scores are adjusted for age (and other demographic factors). To determine the relationship between age and CB_1_ and CB_2_, CB_1_ and CB_2_ expression levels in GM and WM were correlated with the corresponding age of each donor. As depicted in Figure 3, only CB_2_ in GM had a significant relationship with age (*p* < 0.002).

### 3.4. CB_1_ Expression and Localization Are Altered in HAND Brains on ART Compared to NUI

To better understand the alterations in CB_1_ levels in the WM and GM of brains from HAND decedents, we performed immunolabelling for CB_1_ in vibratome sections from the frontal cortex. First, we identified CB_1_ expression in neurons in vibratome sections from the frontal cortex. The signal for CB_1_ was clear and punctate throughout the neuronal processes in the GM in brains of HIV+ NUI decedents, however, CB_1_ signal was very strong in the soma of neurons in the brains of HAND decedents (Figure 4A). Interestingly, the CB_1_ dotting the neuronal processes throughout the GM was less common in the brains of decedents diagnosed with HAND (Figure 4A). Next, to identify CB_1_ co-localization with neurons, we double-immunolabeled vibratome sections with antibodies against CB_1_ and MAP2 and analyzed them using confocal microscopy. In the brains from the decedents diagnosed as NUI, CB_1_ stained strongly throughout the processes of MAP2+ cells (Figure 4B). However, the signal for CB_1_ (red) was much more apparent in the soma of neurons in brains from the decedents diagnosed with HAND when compared to CB_1_ localization in brains from the decedents diagnosed as NUI (Figure 4C). The quantification of the signal for CB_1_ co-localizing with the signal for MAP2 was not significantly different in brains of the decedents diagnosed as HAND when compared to CB_1_ levels in the brains of the decedents diagnosed as NUI (Figure 4C). Next, we visualized CB_1_ immunostaining in cells with glial morphology, which were apparent in WM but not GM. The signal for CB_1_ was apparent in WM glial cells in the brains of HIV+ NUI decedents. However, CB_1_ signal was very strong in glial processes that extend from the soma, more so in the brains from the HAND decedents than those from NUI decedents (Figure 4D). Strongly stained glia cells were present in WM but not in GM. Next, to identify CB_1_+ astroglia, we double-immunolabeled vibratome sections with antibodies against CB_1_ and GFAP, which was visualized using confocal microscopy. In the brains from the decedents diagnosed as NUI, CB_1_ stained lightly, with little colocalization with the GFAP signal (Figure 4E). CB_1_ stained more strongly in the brains from the decedents diagnosed with HAND (Figure 4E). Quantification of the signal for CB_1_ co-localizing with the signal for GFAP increased by ~70% in the brains of decedents diagnosed as HAND when compared to CB_1_ levels in the brains of decedents diagnosed as NUI (Figure 4F). These data suggest that astroglia increase expression of CB_1_ during HAND, while CB_1_ distribution in neurons may be abnormal in HAND, suggesting that astroglial CB1 may be an optimal therapeutic target using cannabinoids.

## 4. Discussion

The current study provides evidence that the ECS is altered in brain tissues of HIV+ decedents that were diagnosed with HAND while on ART. This study is the first to analyze CB_1_ and CB_2_ receptor expression in HAND brains on ART. We identify pathological changes in CB_1_ localization in neurons and astroglia that may reflect an attempt by brain cells to restore neuronal homeostasis. Correlational analyses between CB_1_ levels and specific neurocognitive domains suggest that ECS changes associated with HIV and ART may be related to the development of HAND. These findings are consistent with previous studies that have reported an increase in CB_1_ levels in PWH with HIV encephalitis and may implicate the ECS as a promising therapeutic target for PWH with HAND in the ART era [27] and in neuroinflammatory diseases in general [36].

Several studies have suggested that PWH use cannabis at a higher rate than the general population, with studies showing that 14–56% of PWH using cannabis compared to <10% in the general population [19,37,38,39,40,41]. The higher rates of use among PWH may be due to cannabis’ ability to alleviate anxiety, depression, nausea, sleep disorders, and other symptoms associated with HIV infection [20,42]. While the evidence that cannabis effectively eliminates such symptoms in all patients is variable, our data suggest that the ECS is upregulated in response to HIV infection and ART, possibly as a compensatory mechanism to restore brain homeostasis. Future studies of how cannabis use affects the ECS in the brains of PWH are needed to better understand the therapeutic potential of drugs targeting the ECS.

Alterations in CB_1_ and CB_2_ expression may constitute a compensatory response to neuroinflammation that persists in PWH on ART, despite low or undetectable viral loads [6,7,43]. In particular, increases in WM CB_1_ may compensate for the observed decreases in WM CB_2_ levels. Indeed, chronic inflammation in PWH on ART likely contributes to comorbidities such as HAND and depression. However, the significant relationship between GM CB_2_ protein levels and age suggest that other factors may play a role in CB receptor levels, such as the duration of HIV infection. Surprisingly, we found no significant relationship between CB_1_ and CB_2_ levels and plasma viral load or CD4 count. These findings may suggest that CB_1_ and CB_2_ levels are influenced by low-level HIV protein expression or ART regimens. These hypotheses need to be tested in a larger cohort with more closely age-matched controls. Several studies have suggested that cannabinoids are anti-inflammatory and neuroprotective [10,44,45,46]. Our recent studies showed that a cannabinoid receptor agonist, WIN55,212-2, may be neuroprotective by reducing neuroinflammatory gene expression in reactive astroglia, although we found that WIN55,212-2 was acting through peroxisome proliferator-activated receptors (PPAR) [13,47]. Importantly, there is evidence that the phytocannabinoids isolated from cannabis, ∆9-tetrahydrocannabinol (THC), and cannabidiol (CBD) activate PPAR α and γ via signaling pathways that could be downstream of, and also independent of, CB_1_ and CB_2_ [48,49,50]. Other studies have shown that THC and CBD are neuroprotective in animal models for neurodegenerative diseases [46,51]. Thus, THC and CBD may mimic endocannabinoid signaling mediated down modulation of neuroinflammation, a process that may be perturbed in PWH. These findings warrant further studies to delineate the mechanisms of cannabis-mediated neuroprotection and the role of the ECS in these processes to better understand the mechanisms of action underlying the therapeutic effects.

The neuronal pathology observed in HAND brains compared to NUI may reflect altered distribution of mitochondria throughout neuronal soma and processes. The CB_1_ receptor has been shown to be located at mitochondria as well as on the plasma membrane of cells [52]. CB_1_ has also been reported to be upregulated in neurodegenerative diseases [27]. Cannabinoid receptor agonists have shown promising therapeutic effects in multiple animal models for neurodegenerative diseases. Additionally, previous studies have shown that cannabinoid receptor agonists alter inflammatory gene expression and mitochondrial metabolic processes in multiple cell types [13,20,47,48,52,53,54,55,56,57,58]. Astroglia and neurons are highly involved in endocannabinoid signaling [55,57,59,60]. Moreover, endocannabinoid signaling regulates neurotransmission and metabolism in and between the two brain cell types [12,46,50,59,60,61,62,63]. These studies, for the first time, identified a shift in the localization of CB_1_ receptors on neurons from punctate distribution to localization to the soma of neurons, which may be associated with a shift in mitochondrial fission and fusion processes in neurons, as we reported in HAND brains [33]. Indeed, our findings in NUI brains are consistent with previous findings that showed that a subset of CB_1_ receptors dot the outer membranes of mitochondria, where they alter the function of the electron transport chain among other pathways [54,56]. However, more studies are needed to confirm this hypothesis. In the least, the alterations in CB_1_ and CB_2_ expression and localization in HAND may indicate the ECS as a promising target for therapeutic intervention. This assertion is consistent with recent studies showing that PWH who use cannabis may demonstrate improved cognitive function compared to PWH not using cannabis [45,64]. Moreover, other neurodegenerative processes, including traumatic brain injury, AD, MS, and HD are associated with altered levels of ECS components, suggesting the modulation of the ECS as a potential therapeutic strategy [22,23,24,65]. Future studies are needed to develop therapies aimed at astroglia to restore immunometabolic balance in the brain.

While important, relevant, and timely observations are presented, this study has several noteworthy limitations. A potential caveat in this study was the limited cohort size, which is probably not fully representative of the whole population of PWH. Future studies using brain specimens from a larger cohort that is demographically representative of PWH in regard to race, sex, age, and other characteristics may more accurately elucidate the effects of HIV on CB_1_ and CB_2_ expression in the brain. Including HIV controls in future analyses is important to understand how HIV infection affects CB_1_ and CB_2_ expression independent of individuals with HAND status. Important specimen data on the duration of HIV infection, duration of ART, recency and frequency of exposure to cannabis and other drugs of abuse were not available for all participants and therefore limited the study to only correlations with HIV clinical data and the neuropsychological measures of HAND. The analyses here are limited to only frontal cortex tissues, while CB_1_ and CB_2_ are expressed throughout the brain and in regions such as the hippocampus and striatum, which are implicated to be involved in HAND phenotypes. CB_1_ and CB_2_ are expressed in brain cell-types other than astroglia and neurons, including microglia and possibly oligodendroglia, and it is important to perform future studies that determine CB_1_ and CB_2_ levels in these cell types in specimens from PWH with HAND on ART. The ECS is composed of endocannabinoid-producing and -degrading enzymes, endocannabinoids, as well as receptors aside from CB_1_ and CB_2_, none of which were examined in this investigation. CB_1_ and CB_2_ expression levels could not be related to markers of neuroinflammation or neurodegeneration, as these data are currently unavailable for a majority of the specimens; future studies, therefore, are necessary to assess these correlations. Psychiatric factors, such as depression, may influence neurocognitive impairment, and they should be considered in any future studies. Finally, the data presented here are observational and associative, lacking mechanistic links between HIV, ART, and the changes in CB_1_ and CB_2_. This work provides support for future studies to identify HIV and ART-associated factors leading to alterations in CB_1_ and CB_2_ in the brain, as well as a better understanding of the potential ameliorative role of ECS targeting therapies for PWH and other neurodegenerative diseases.

## 5. Conclusions

Overall, this study identified alterations in CB_1_ and CB_2_ expression as potential mechanisms contributing to, or, alternatively, a consequence of, the neuropathogenesis of HAND during the ART era. Importantly, these shifts in CB_1_ and CB_2_ expression may be dependent on cell-types and therefore could indicate that therapies designed to target these receptors or other ECS proteins in neurons or glial cells may be beneficial for patients with HAND. These findings further support the recent interest in astroglia as mediators of neurodegenerative diseases and studies that show promising effects of cannabinoids in combatting such diseases [59,66]. Future studies are needed to develop therapies aimed at astroglia and neurons to restore brain homeostasis.

## Figures and Tables

**Figure 1 viruses-13-01742-f001:**
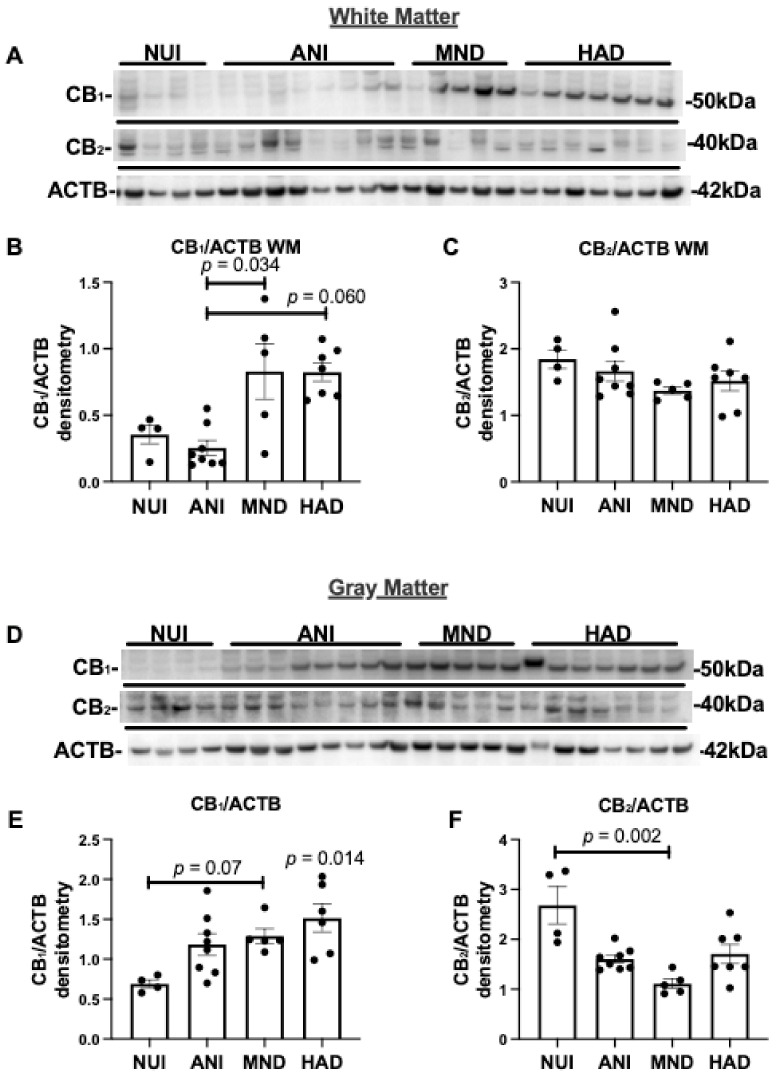
Cannabinoid (CB)_1_ and CB_2_ expression are increased in HIV associated neurocognitive disorder (HAND) brains on antiretroviral therapy (ART): (**A**) Immunoblot of HAND donor white matter (WM) frontal lobe (Brodmann area 46) lysates with antibodies specific for CB_1_ and CB_2_, and β-actin ACTB. (**B**,**C**) Quantification of normalized band intensity of CB_1_ and CB_2_ in WM stratified by HAND diagnosis. (**D**) Immunoblot of HAND donor gray matter (GM) frontal lobe lysates with antibodies specific for CB_1_ and CB_2_, and ACTB. (**E**,**F**) Quantification of normalized band intensity of CB_1_ and CB_2_ in GM stratified by HAND diagnosis. Samples were run on immunoblot in the same order as listed in Table 1. Statistical significance was determined by Kruskal–Wallis test; Error bars represent standard error of the mean.

**Figure 2 viruses-13-01742-f002:**
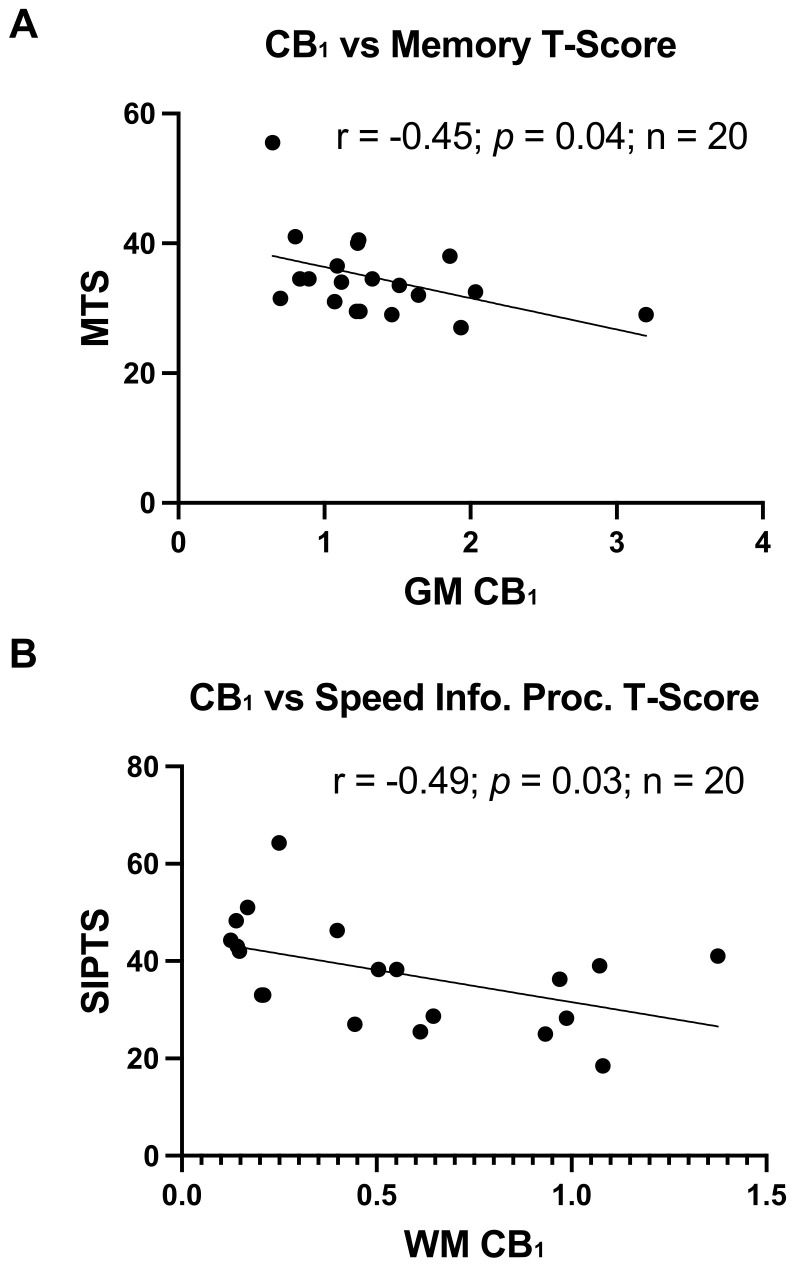
Greater CB_1_ expression relates to worse cognitive function: (**A**) Correlation of CB_1_ expression in GM with memory T-scores, r = −0.45, *p* = 0.04. (**B**) Correlation of CB_1_ expression in WM detected by immunoblot with speed of information processing T-scores, r = −0.49, *p* = 0.03.

**Figure 3 viruses-13-01742-f003:**
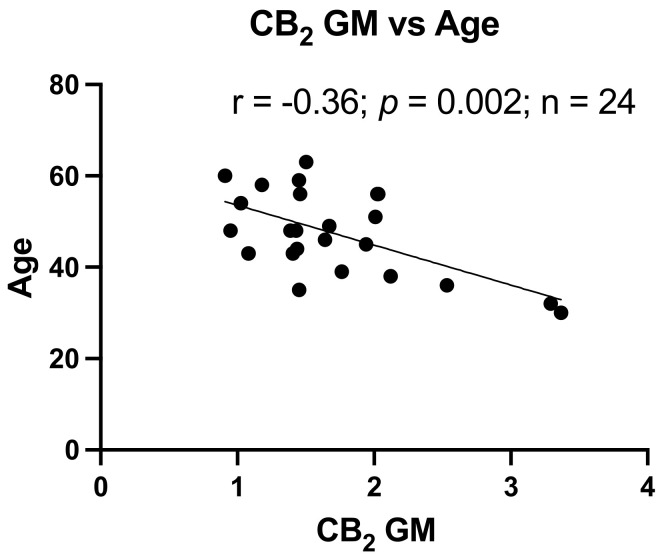
Reduced CB_2_ expression relates to increased age. Correlation of CB_2_ expression in GM with Age, r = −0.36, *p* = 0.002.

**Figure 4 viruses-13-01742-f004:**
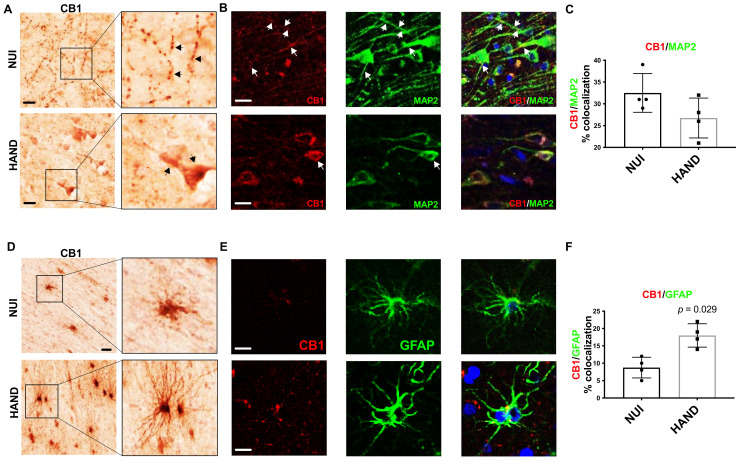
CB_1_ expression and localization is altered in HAND brains on ART compared to NUI: (**A**) Immunostaining of CB_1_ in the frontal cortices of NUI and HAND brain tissue. (**B**) Double-immunostaining of CB_1_ and MAP2 in frontal cortices of HAND and NUI brain tissue. (**C**) CB_1_ signal %colocalization of CB_1_/MAP2. (**D**) Immunostaining of CB_1_ in the frontal cortices of NUI and HAND brain tissue and the quantification of CB_1_+ glia. (**E**) Double-immunostaining of CB_1_ and GFAP in frontal cortices of HAND and NUI brain tissue. (**F**) Quantification of %colocalization of CB_1_/GFAP. Statistical significance was determined by Mann–Whitney test; Error bars represent standard error of the mean.

**Table 1 viruses-13-01742-t001:** Clinical characteristics of study population. Abbreviations: NUI, Neurocognitively unimpaired; ANI, Asymptomatic neurocognitive impairment; MND, Minor neurocognitive dysfunction; HAD, HIV-associated dementia; CD4, cluster of differentiation 4.

Neurocognitive Diagnoses	Age	Plasma Viral Load	CD4 Count	ART Ever Taken
NUI	45	41,700	N/A	3TC
NUI	30	40	72	ATR
NUI	32	400	54	CBV/FTV/RTV/SQV/TRU
NUI	38	29,282	663	3TC/ABC/ATV/DDI/KTA/RTV/TFV/ZDV
ANI	63	50	516	3TC/ATR/CBV/D4T/EFV/EPZ/IDV/NFV/TFV/TRU/ZDV/DRV/RTV/RPV/ATV
ANI	49	400	1	3TC/D4T/DDI/EFV/FTC/FTV/KTA/NFV/T20/TFV/TRU
ANI	46	750,000	18	3TC/ABC/APV/D4T/EFV/NFV/ZDV/CBV/TZV
ANI	48	9501	80	3TC/D4T/DDC/DDI/EFV/HU/IDV/NFV/NVP/RTV/SQV/ZDV
ANI	43	5722	63	3TC/CBV/D4T/IDV/NFV/NVP/SQV
ANI	56	750,000	70	3TC/D4T/DDI/EFV/NFV/NVP/ZDV/KTA/RTV/TFV/CBV/SQV/TRU/FPV
ANI	39	1013	402	3TC/D4T/IDV/TZV
ANI	48	50	480	ABC/ATV/DLV/DRV/MVC/RTV/TFV/TMC
MND	58	4064	14	ATR/ATV/DRV/FTC/KTA/RGV/RTV/TFV/TRU/TZV
MND	44	133,166	7	CBV/KTA/TFV
MND	48	53,556	77	CBV/EFV/NFV/ZDV
MND	60	50	119	3TC/EFV/HU/TFV
MND	43	50	69	3TC/ABC/D4T/DLV/EFV/IDV/NFV/ZDV/CBV/DDI
HAD	56	61,223	24	3TC/ABC/ATV/D4T/DDI/EFV/EPZ/FTV/IDV/KTA/NFV/NVP/RTV/T20/3TC
HAD	54	400	336	ATV/CBV/FTC/KTA/NVP/RTV/TFV/TRU
HAD	59	400	32	3TC/DDC/DLV/EFV/KTA/TFV/ZDV
HAD	36	6952	63	D4T/3TC/IDV/NVP/RTV/ABC/KTA/NFV/DDI
HAD	51	605,555	34	TFV/FTC
HAD	56	1631	8	CBV/NVP
HAD	35	85,510	3	ABC/CBV/D4T/DDI/IDV/NVP/RTV/SQV/ZDV/CBV

**Table 2 viruses-13-01742-t002:** Mean of clinical characteristics: Mean and standard deviation of age, plasma viral load, and CD4+ cell count were calculated for the four groups: NUI, ANI, MND, and HAD.

Diagnoses	Age	Plasma Viral Load	CD4 Count
NUI	36.25 ± 6.78	17,855.5 ± 20,985.8	263 ± 346.5
ANI	49 ± 7.5	189,592 ± 345,907.3	203.8 ± 220.9
MND	50.6 ± 7.9	38,177.2 ± 57,728.8	57.2 ± 46.7
HAD	49.6 ± 9.9	108,810.1 ± 221,709.6	71.4 ± 118.3

## Data Availability

All data will be available by reasonable request and will also be deposited with the National NeuroAIDS Tissue Consortium.

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
