# Peer review of "Alterations in Brain Cannabinoid Receptor Levels Are Associated with HIV-Associated Neurocognitive Disorders in the ART Era: Implications for Therapeutic Strategies Targeting the Endocannabinoid System"

_viruses, 2021, doi:10.3390/v13091742_

Round 1

Reviewer 1 Report

  1. Please check line 109, page no. 3 – “Tissues from the white matter (GM) and gray matter WM of the frontal cortices dissected”. The short forms seem to be incorrect. Please correct.
  2. It would be better if the authors would have provided full forms of ART taken in the supplementary material. It will help readers to understand the paper better way.
  3. Have authors checked the inflammatory markers in HIV+ brain specimens?

Author Response

  1. Please check line 109, page no. 3 – “Tissues from the white matter (GM) and gray matter WM of the frontal cortices dissected”. The short forms seem to be incorrect. Please correct.

Response: Thank you. This was corrected in the revised manuscript.

  1. It would be better if the authors would have provided full forms of ART taken in the supplementary material. It will help readers to understand the paper better way.

Response: A list of ART abbreviations was added to the manuscript, placed before the References.

  1. Have authors checked the inflammatory markers in HIV+ brain specimens?

Response: This is a great question. We have not yet analyzed levels of inflammatory cytokines in all of the brain specimens. However, we will include this readout in future analyses and we mentioned this in the Discussion of the revised manuscript on page 12, lines 442-444.

“CB1 and CB2 expression levels could not be related to markers of neuroinflammation or neurodegeneration as these data are currently unavailable; future studies are necessary to assess these correlations.”

Reviewer 2 Report

This manuscript describes alterations in cannabinoid receptor expression and localization in individuals with HIV and differing levels of cognitive impairment. Cognitive impairment is a serious complication of HIV, and this study reports important findings about the nature of CB1 (and to a lesser extent) CB2 receptor expression. The manuscript is well-written and the introduction provides a strong basis for the study. Although limited in sample size and overall scope, this work is novel, and correlations with the degree of cognitive impairment make the manuscript particularly impactful. Concerns are listed below:

  • State whether cannabis use was known for subjects
  • Table 1: define abbreviations for ARTs
  • Fig 1: define acronyms in caption
  • Fig 1b: data do not appear to meet assumptions of ANOVA
  • Fig 1: define what error bars represent
  • Provide more details regarding the immunohistochemistry: region of frontal cortex analyzed, number of fields imaged per brain, area of each image, etc.
  • Results: state exact p values and effect sizes throughout
  • Methods: Provide more details for brightfield microscopy

Author Response

  • State whether cannabis use was known for subjects

Response: This was added to the discussion: page 12, lines 430-433“.

“Important specimen data on duration of HIV infection, duration on ART, recency and frequency of exposure to cannabis and other drugs of abuse were not available for all participants and therefore limited the study to only correlations with HIV clinical data and neuropsychological measures of HAND.”

  • Table 1: define abbreviations for ARTs

Response: This was added to the manuscript on page 13.

  • Fig 1: define acronyms in caption

Response: These were added.

  • Fig 1b: data do not appear to meet assumptions of ANOVA

Response: We agree that the cohort size is small. We analyzed the data using the Kruskal-Wallis test (non-parametric); p-values were not as robust, but significant differences remained. These findings need to be confirmed in future studies using a larger and more well-characterized cohort. However, we feel these data are useful and timely to the field.

  • Fig 1: define what error bars represent

Response: The error bars represent standard error of the mean. This was added to figure captions.

  • Provide more details regarding the immunohistochemistry: region of frontal cortex analyzed, number of fields imaged per brain, area of each image, etc.

Response: These details were added to the methods section: page 5, lines 188-190.

“a total of 3 sections (10 images per section at 200x magnification) were analyzed in order to quantify the average number of immunolabelled cells per field of view (100µm2).”

  • Results: state exact p values and effect sizes throughout

Response: These data were added to the graphs

  • Methods: Provide more details for brightfield microscopy

Response: These details were added to the methods section: page 5, lines 170-172.

“a total of 3 sections (10 images per section at 200x magnification) were analyzed in order to quantify the average number of immunolabelled cells per field of view (400µm2).”

Reviewer 3 Report

This is a well written article on an important subject. The results seem to support the conclusions.

A few points regarding the writing and background need to be improved.

  1. Depending on the journal policy, its is standard (and useful) to define important terms used in the abstract even if they are defined in the text. So consider defining HAND ART and PWH in the abstract.
  2. Be sure to be consistent with spaces before references. Example - line 58: '...mood problems[20, 21]...' should read '...mood problems [20, 21]...' Many other examples.
  3. The introduction of a manuscript on cannabinoid receptors should define the class of receptors, thus, at line 61, insert:

'The cannabinoid CB1 and CB2 receptors are G protein coupled receptors [X].'

X.  Miles D Thompson, Takeshi Sakurai, Innocenzo Rainero, Mary C Maj, Jyrki P Kukkonen. Orexin Receptor Multimerization versus Functional Interactions: Neuropharmacological Implications for Opioid and Cannabinoid Signaling and Pharmacogenetics. Pharmaceuticals (Basel) . 2017 Oct 8;10(4):79. doi: 10.3390/ph10040079.

Author Response

  1. Depending on the journal policy, it is standard (and useful) to define important terms used in the abstract even if they are defined in the text. So consider defining HAND ART and PWH in the abstract.

Response: Thank you. We have defined the acronyms in the abstract and the changes are in blue font.

  1. Be sure to be consistent with spaces before references. Example - line 58: '...mood problems[20, 21]...' should read '...mood problems [20, 21]...' Many other examples.

Response: Thank you. We have placed a space before all citations.

  1. The introduction of a manuscript on cannabinoid receptors should define the class of receptors, thus, at line 61, insert:

'The cannabinoid CB1 and CB2 receptors are G protein coupled receptors [X].'

  1. Miles D Thompson, Takeshi Sakurai, Innocenzo Rainero, Mary C Maj, Jyrki P Kukkonen. Orexin Receptor Multimerization versus Functional Interactions: Neuropharmacological Implications for Opioid and Cannabinoid Signaling and Pharmacogenetics. Pharmaceuticals (Basel) . 2017 Oct 8;10(4):79. doi: 10.3390/ph10040079.

Response: Thank you. This has been to the revised version on page 2, lines 64-65.

“Cannabinoid receptors (CB) CB1 and CB2 are G protein coupled receptors [26].”

Reviewer 4 Report

The main observation presented by Authors is that CB1/CB2 expression levels elevate in HIV-infected patients that developed HAND and that localization of CB1/CB2 changes upon HAND. Based on this they formed a conclusion that CB1/CB2 system could be a target for anti-HAND pharmacotherapy.

“Rates of cannabis (i.e. marijuana) use are disproportionately high in this population with approximately 77% of HIV-infected adults reporting lifetime marijuana use
compared to 44.5% in uninfected adults [17-19].” - does it mean that more than 40% of adults all over the world use marijuana in their daily life? I doubt such statistics, please clarify.
What is more, this statistics is from Shiau et al. 2017. Yet, it included 377,787 participants from 560,099 total number of participants. The remaining 176,879 participants (one-third of all participants) were excluded from this analysis because they haven’t been checked if they are HIV-positive. Yet, if these 176,879 participants were included in the analysis only one-third of participants could report marijuana use, not 44.5%. Also, there is an age bias. 178024 participants (nearly half) were age 18-25 while most HIV-infected people were age 35-49.
Based on the above, Authors should just mention that marijuana use is significantly higher among HIV-infected people without providing misleading statistics. But still, it does not prove at all or even suggests that CB1/CB2 are good targets for anti-HAND therapy.

Participants included in Table 1 - it would be good to add their average age. Could Authors discuss the impact of participant age on their results? NUI and HAND patients differ by their age. The longer period of suffering from the disease causes neurodegeneration due to, e.g., stress, social exclusion, life-threaten situation, depression. It is not necessarily associated with a direct influence of virus on protein expression levels in astroglia. What is more, Authors stated in lines 312-314 that plasma virus load is not a factor associated with CB1 expression levels.

Correlation -0.5 is rather low to drive conclusions based on this.

Time of living with HIV, or knowing about the infection, should be included in Table 1.

Could Authors exclude participants of older age or at least provide additional results but only from younger patients?

A thorough psychiatric narrative summary should be included for every patient from Table 1 and it could reveal additional factors causing neurocognitive impairment. There should be more factors added apart from age and viral load in Table 1 that could have an impact on results presented in Figure 1.

Are immunoblots in Figure 1A in the same order as patients in Table 1? Include it in the Figure caption. Add a description to the CD4 abbreviation. Is there any correlation between CB2 levels and plasma viral load? Please, comment on this.

Line 269 - Astrocytes have a well-defined role in neuroprotection, so it’s not a novel conclusion.

ECS-based pharmacotherapy may be good for HIV-infected patients due to pain release, especially in terminal stages. Yet, forming a conclusion that they should use marijuana to prevent HAND is an exaggeration and may cause an additional, unwanted substance abuse state.

Instead of focusing on HIV-infected patients, Authors should refer to other diseases causing neurocognitive disorders, e.g. traumatic brain injury (Vogel et al. Sci Rep 2020). The manuscript lacks any reference to ECS-based therapies in such disorders. Authors should also test endocannabinoids levels. Are they low?

Author Response

  1. The main observation presented by Authors is that CB1/CB2 expression levels elevate in HIV-infected patients that developed HAND and that localization of CB1/CB2 changes upon HAND. Based on this they formed a conclusion that CB1/CB2 system could be a target for anti-HAND pharmacotherapy. What is more, this statistics is from Shiau et al. 2017. Yet, it included 377,787 participants from 560,099 total number of participants. The remaining 176,879 participants (one-third of all participants) were excluded from this analysis because they haven’t been checked if they are HIV-positive. Yet, if these 176,879 participants were included in the analysis only one-third of participants could report marijuana use, not 44.5%. Also, there is an age bias. 178024 participants (nearly half) were age 18-25 while most HIV-infected people were age 35-49.

“Rates of cannabis (i.e. marijuana) use are disproportionately high in this population with approximately 77% of HIV-infected adults reporting lifetime marijuana use
compared to 44.5% in uninfected adults [17-19].” - does it mean that more than 40% of adults all over the world use marijuana in their daily life? I doubt such statistics, please clarify.

Response: This is a good point. The language has been edited to state that this percentage of people report using at least once in their lifetime. Page 2, lines 54-58.

“Rates of cannabis (i.e. marijuana) use are disproportionately high in this population with some studies showing as high as 77% of HIV-infected adults reporting marijuana use at least once in their lifetime compared to 44.5% in uninfected adults [17-19]. However, in the case of Shiau et al. [18], as with many surveys of a large cohort, the data may be subject to selection bias that skew the results.”

  1. Based on the above, Authors should just mention that marijuana use is significantly higher among HIV-infected people without providing misleading statistics. But still, it does not prove at all or even suggests that CB1/CB2 are good targets for anti-HAND therapy.

Response: The language was edited to clearly state the statistics and limitations of such statistics. Page 2, lines 54-58.

“Rates of cannabis (i.e. marijuana) use are disproportionately high in this population with some studies showing as high as 77% of HIV-infected adults reporting marijuana use at least once in their lifetime compared to 44.5% in uninfected adults [17-19]. However, in the case of Shiau et al. [18], as with many surveys of a large cohort, the data may be subject to selection bias that skew the results.”

  1. Participants included in Table 1 - it would be good to add their average age. Could Authors discuss the impact of participant age on their results? NUI and HAND patients differ by their age. The longer period of suffering from the disease causes neurodegeneration due to, e.g., stress, social exclusion, life-threaten situation, depression. It is not necessarily associated with a direct influence of virus on protein expression levels in astroglia. What is more, Authors stated in lines 312-314 that plasma virus load is not a factor associated with CB1 expression levels.

Response: These are very good suggestions. We’ve added a second table (Table 2) to the text that includes average age, CD4 count, and plasma viral load for each group. We also ran simple linear regression analyses between CB1/2 protein levels in GM and WM with age, CD4 count, and plasma viral load. Only age and CB2 GM showed a significant relationship and this plot was added as Figure 3. We also added the results and discussion to the revised version. We noted that the relationships between CB1/2 levels and viral load and CD4 count were not significant. Page 11, lines 377-383.

“However, the significant relationship between GM CB2 protein levels and age suggests that other factors may play a role in CB receptor levels. Surprisingly, we found no significant relationship between CB1 and CB2 levels and plasma viral load or CD4 count. These findings may suggest that CB1 and CB2 levels are influenced by low-level HIV protein expression or ART regimens. These hypotheses need to be tested in a larger cohort with better age-matched controls.”

Moreover, we added to page 8 lines 287-289 that “neurocognitive T-scores are adjusted for age (and other demographic factors).”

  1. Correlation -0.5 is rather low to drive conclusions based on this.

Response: According to Cohen (1988, 1992), the effect sizes for Pearson Correlation Coefficients are as follows:  R≈0.1 is small, R≈0.3 is medium, R≈0.5 is large. Accordingly, this is a rather large correlation that we believe supports the hypothesis that CB levels are associated with neurocognitive outcomes. However, we think it is important for these results to be replicated in a larger sample and we state this in the Discussion.

  1. Time of living with HIV, or knowing about the infection, should be included in Table 1.

Response: We agree that this is important information. However, accurate data for these measures are not available. Many of the patients do not know when they were initially infected with HIV. Moreover, other patients made visits to the clinic intermittently and their duration on ART is not fully known. Therefore, rather than provide inaccurate data, we report the drugs that we know the patients taking.

  1. Could Authors exclude participants of older age or at least provide additional results but only from younger patients?

Response: Given the limited number of samples, this further partitioning of the samples by age will prevent any meaningful statistical analyses. As stated above, this study needs to be confirmed in a larger number of samples. This was stated in the Discussion section (page 12, lines 444-446).

“Future studies using brain specimens from a larger cohort that is demographically representative of PWH in regard to race, sex, age and other characteristics may more accurately elucidate the effects of HIV on CB1 and CB2 expression in the brain. Including HIV- controls in future analyses will be important to understand how HIV infection affects CB1and CB2 expression independent of HAND status.”

  1. A thorough psychiatric narrative summary should be included for every patient from Table 1 and it could reveal additional factors causing neurocognitive impairment. There should be more factors added apart from age and viral load in Table 1 that could have an impact on results presented in Figure 1.

Response: These samples were acquired from the National NeuroAIDS Tissue Consortium and no such narrative is available to us. However, it is likely that other psychiatric factors may influence neurocognitive impairment. We included this in the Discussion on page 12, lines 445-446.

“Psychiatric factors, such as depression, may influence neurocognitive impairment and they should be considered in any future studies.”

  1. Are immunoblots in Figure 1A in the same order as patients in Table 1? Include it in the Figure caption. Add a description to the CD4 abbreviation.

Response: We added these details to the figure 1 caption.

  1. Is there any correlation between CB2 levels and plasma viral load? Please, comment on this.

Response: No, there is no significant association between CB2 levels and viral load. This was added to the Discussion section: page 10, line 359-364.

  1. ECS-based pharmacotherapy may be good for HIV-infected patients due to pain release, especially in terminal stages. Yet, forming a conclusion that they should use marijuana to prevent HAND is an exaggeration and may cause an additional, unwanted substance abuse state.

Response: We agree with the reviewer and we have edited the Discussion accordingly. While we do see evidence in the literature that modulation of the endocannabinoid system may be therapeutic for neurodegenerative diseases, we do not intend to promote self-medication with cannabis.

The second paragraph of the Discussion (page 11, lines 370-373) now states:

“Future studies of how cannabis use affects the ECS in the brain in PWH are needed to better understand the therapeutic potential of drugs targeting the ECS.”

The fifth paragraph of the Discussion (page 12, lines 448-451) now states:

“This work provides support for future studies to identify HIV and ART-associated factors leading to alterations in CB1and CB2 in the brain as well as a better understanding of the potential ameliorative role of ECS targeting therapies in PWH and other neurodegenerative diseases.”

  1. Instead of focusing on HIV-infected patients, Authors should refer to other diseases causing neurocognitive disorders, e.g. traumatic brain injury (Vogel et al. Sci Rep 2020). The manuscript lacks any reference to ECS-based therapies in such disorders.

Response: This is a good suggestion. We edited the text to discuss how cannabinoids are being developed to treat other neurodegenerative diseases. Page 2, lines 60-64.

“Cannabis may have promising utility in treating various neurodegenerative disorders with underlying inflammatory processes, and the expression of cannabinoid receptors is associated with Alzheimer’s disease [22], multiple sclerosis [23], Huntington’s disease [24], and Down’s syndrome [25].”

And on page  11 lines 418-420:

“Moreover, other neurodegenerative processes, including traumatic brain injury, AD, MS, and HD are associated with altered levels of ECS components, suggesting the modulation of the ECS as a potential therapeutic strategy [22 ,23, 24, 65].”

  1. Authors should also test endocannabinoids levels. Are they low?

Response: This is a good suggestion, but unfortunately beyond our budget and thus out of the scope of this study. We added this as a limitation of the study, page12, lines 440-442.

“The ECS is composed of endocannabinoid-producing and -degrading enzymes, endocannabinoids, as well as receptors aside from CB1 and CB2; none of which were examined in this investigation.”

Round 2

Reviewer 2 Report

The article is much improved following revision. My only minor concern is that the field of view for microscopy is described as 100µmin one instance, but 400 µm2 in another instance. Please correct if this is a mistake or explain the discrepancy in the text.

Reviewer 4 Report

Authors responded to most of the comments.